# OmniMixup: Generalize Mixup with Mixing-Pair Sampling Distribution

## Abstract

Mixup is a widely-adopted data augmentation techniques to mitigates the over-fitting issue in empirical risk minimization. Current works of modifying Mixup are modality-specific, thereby limiting the applicability across diverse modalities. Although alternative approaches try circumventing such barrier via mixing-up data from latent features based on sampling distribution, they still require domain knowledge for designing sampling distribution. Moreover, a unified theoretical framework for analyzing the generalization bound for this line of research remains absent. In this paper, we introduce OmniMixup, a generalization of prior works by introducing Mixing-Pair Sampling Distribution (MPSD), accompanied by a holistic theoretical analysis framwork. We find both theoretically and empirically that the Mahalanobis distance (M-Score), derived from the sampling distribution, offers significant insights into OmniMixup's generalization capabilities. Accordingly, we propose OmniEval, an evaluation framework designed to autonomously identify the optimal sampling distribution. The empirical study on both images and molecules demonstrates that 1) OmniEval is adept at determining the appropriate sampling distribution for OmniMixup, and 2) OmniMixup exhibits promising capability for application across various modalities and domains.

## 1 Introduction

By creating virtual data from a pair of training samples, Mixup (Zhang et al., 2017; Tokozume et al., 2018; Zhang et al., 2020) has been shown to bolster the robustness and generalization capacity of models, yielding non-trivial improvements on various domains, such as image classification (Yun et al., 2019; Kim et al., 2020; Hong et al., 2021), and Natural Language Processing (NLP) (Yoon et al., 2021; Kong et al., 2022; Guo et al.; Sun et al., 2020). Conventionally, Mixup is conducted in input-level, which requires konwledge of data structure in order to delicately design mixup strategy in sub-data level e.g., image patches (Faramarzi et al., 2022), word tokens (Yoon et al., 2021) , to reassemble to new samples. However, such technique tend to be specific to certain modalities or domains, which constrains the broader application of Mixup. The quest for a universally effective mixup method that accommodates diverse data modalities remains intriguing for the communities.

Accordingly, recent advances proposed to mixing feature instead of the input data (Verma et al., 2019; Faramarzi et al., 2022; Baena et al., 2022), as data with different modalities and dimensions can be project into a unified and shared latent space. This line of research primarily focus on circumventing the so called *manifold intrusion* and the corresponding modifications can be categorized into three major directions: 1) modifications to the hidden states used for mixup (Verma et al., 2019; Faramarzi et al., 2022); 2) adjustments to the sampling of the mixup ratio $\lambda$ (Guo et al., 2019); 3) modifications to the data sampling distribution (Baena et al., 2022; Yao et al., 2022; Hwang & Whang, 2021), which stands as the primary focus of this work. Currently, the key idea of this line of research is to mixup samples based on similarity, thereby preventing erroneous augmented data caused by out-of-distribution virtual samples. For example, Local-Mix (Baena et al., 2022) and C-Mixup (Yao et al., 2022) suggest to mixup samples based on data or label similarity, respectively. However, the application of current approaches suffers from the following limitations: 1) the design of sampling distribution is based on similarity, which still necessitates domain knowledge; 2) an holistic theoretical framework for comparing generalization ability of different approaches across varied domains and modalities remains unexplored.

As a remedy for solving both limitations together, this paper introduces **OmniMixup**, a mixup framework that is capable of encompassing all the related prior works by introducing Mixup-Pair Sampling Distribution (MPSD), and present a theoretical framework to analyze the generalization ability of these works all together. Previous work (Zhang et al., 2020) first proposes a theoretical analysis on the vanilla mixup (Zhang et al., 2017) toward the relation of generalization and the intrinsic dimension of dataset, while the question toward various advanced mixup strategies remains unsolved. Park et al. (2022a) further provides an unified theoretical framework for Mixup and Cut-Mix, yet their application remained tethered to image data. In this work, we theoretically analyze the effectiveness of sampling distribution in feature-based mixup approaches, thereby providing a valuable insight for the broader application of Mixup. Buliding upon Zhang et al. (2020)'s foundation, we find that the expectation of the Mahalanobis distance (**M-Score**) within MPSDs is informative to the generalization ability of OmniMixup. Guided by this discovery, we provide a efficient while effective evaluation framework, **OmniEval**, for evaluating the effectivness of MPSDs based on information within the M-Score. OmniEval allow us to identify the MPSD that will lead to a strong performance of the resulting model. To achieve this, OmniEval overcomes two challenges when evaluating MPSDs with M-Score. First, it obviates the need for expensive model training with every MPSD, requiring only a model trained via ERM; second, OmniEval estimate the M-Score to circumvent the intractable nature of the calculation of the expected M-Score.

Overall, the contribution of this paper can be summarized as follows:

1. We propose OmniMixup to generalize the vanilla Mixup with arbitrary sampling distribution and provide an holistic theoretical framework toward their ability in generalization.

2. We present OmniEval based on estimating the expected M-Score under ERM setting to identify an appropriate MPSD for training models with OmniMixup.

3. We conduct experiments on image classification and molecular property prediction to both verify the effectiveness and transferability of the proposed framework.

## 2 RELATED WORK

### 2.1 MIXUP

Mixup is a commonly used data augmentation technique, especially in the field of computer vision and NLP. Zhang et al. (2020) and Tokozume et al. (2018) first proposed to interpolate training samples linearly to conduct new augmented samples to address the overfitting issue in empirical risk minimization. Currently, there are two strands of mixup research works. The predominant approach encompasses structure-based mixup methods, wherein samples are mixed before being fed into neural networks. For example, Guo et al. (2019); Yun et al. (2019); Faramarzi et al. (2022); Beckham et al. (2019); Summers & Dinneen (2019); Hong et al. (2021) proposed mixup strategies to mix two or more images together to generate new training data, Guo & Mao (2021); Han et al. (2022); Park et al. (2022b); Navarro & Segarra (2023) edit graph topologically (i.e. modify nodes and edges) to mix different graphs together. This line of research have stronger performance due to the fact that it incorporates more domain prior knowledge in the mixup strategy. Moreorever, it is also modality-specific, which subjects the ability of generalization of the mixup strategy. For example, mixup strategies for images cannot be used in graph data, and vice versa.

Cocurrently, another line of research focus on mixing the latent features of data. For example, ManifoldMixup (Verma et al., 2019) proposed to mixup the features in each layer of the deep neural network to foster smoother decision boundary for classifiers; NFM (Lim et al., 2021) proposes to add noises before mixing up; k-Mixup (Greenewald et al., 2021) proposed to mixup $k$ samples to avoid generating points with wrong labels when the data manifold is complicated. This research aligns with our focus, wherein we endeavor to generalize the vanilla mixup (Zhang et al., 2017) from MPSD, and provide analysis both theoretically and empirically.

In contrast to the previous work (Baena et al., 2022; Yao et al., 2022) which focus on the design of MPSD aiming to address the manifold intrusion issue, or improve the robustness of the models, in this paper, we revisit all of these methods and propose a generalized version of Mixup to summarize all these methods. This allows us to analyze all these methods in a unified theoretical framework. Furthermore, the proposed OmniEval framework in the paper can help automously identify the

appropriate MPSD from all of these proposed methods, thereby requiring no domain knowledge in applying OmniMixup in diverse modalities and domains.

## 2.2 MIXUP IN MODELING MOLECULES

The application of deep learning achieves significant improvement in modeling molecules. However, compared to images and text, annotating a molecule is more expensive, as generally it take hours to use DFT to calculate the ground truth label. Data augmentation therefore plays an important role in modeling the molecules (Nakata & Shimazaki, 2017). Although recent advances have focused on mixup approaches for graph data, most of them modify the structure of graph data to generate new examples. This may be unacceptable for molecules, as a slight modification in atom or bond may lead to drastic changes in its chemical properties, thereby making mixup labels to be misleading. However, current approach to apply mixup on molecular data is still from feature levels (Wang et al., 2021). This paper aims at provide an advanced solution of applying mixup for such situation. Specifically, the proposed OmniMixup and OmniEval proposed in this paper can help find an appropriate MPSD automatically and gain improvement without prior domain konwledge for modeling molecules.

## 3 METHODOLOGY

### 3.1 PRELIMINARY

In this sub-section, we introduce the notation in our paper, and present preliminary of Empirical Risk Minimization (ERM) and the vanilla Mixup (Zhang et al., 2017).

**Notations.** A training dataset is denoted as $\mathcal{S} = \{z_1, ..., z_n\}$, where $z_i = (\boldsymbol{x}_i, y_i) \overset{i.i.d.}{\sim} \mathcal{P}_{\boldsymbol{x},y}$, $\boldsymbol{x}_i \in \mathcal{X} \subseteq \mathbb{R}^p$ and $y_i \in \mathcal{Y} \subseteq \mathbb{R}$. The mixed sample of $z_i, z_j$ is denoted as $\check{z}_{i,j}(\lambda) = (\text{mix}(x_i, x_j, \lambda), \text{mix}(y_i, y_j, \lambda))$, where $\text{mix}(a, b, \lambda) = \lambda a + (1 - \lambda)b$. Note that $\lambda \in [0, 1]$. Following Zhang et al. (2020), we denote the mixture distribution of two distribution $\mathcal{D}_1, \mathcal{D}_2$ is denoted as $p\mathcal{D}_1 + (1-p)\mathcal{D}_2$, which suggests that the sample is drawn from $\mathcal{D}_1$ with probability $p$, and $1 - p$ for the another. We denote a model with parameter $\boldsymbol{\theta}$ as $y = f_{\boldsymbol{\theta}}(x)$. We denote $\mathcal{D}_{\boldsymbol{x}}$ as the uniform distribution over $\mathcal{X}$.

**Empirical Risk Minimization.** Under supervised learning, we aim to find a function $f$ such that it can predict labels well given an input. Given an dataset $\mathcal{S} = \{(x_i, y_i)\}_{i=1}^n$, where each datapoint within the dataset is assumed to be i.i.d. sampled from distribution $\mathcal{P}_{x,y}$, and $x \in \mathcal{X}, y \in \mathcal{Y}$. Therefore, the goal is to learn a mapping from $f : \mathcal{X} \to \mathcal{Y}$. Generally, to better help finding such function, a loss function is defined as a mapping of $\ell : \mathcal{Y} \times \mathcal{Y} \to \mathbb{R}$ to evaluate the $f$. A better $f$ will generally lead to a smaller loss function. Based on loss function, the *population risk* (or *expected risk*) is defined as follows:

$$L(f) = \mathbb{E}_{(x,y)\sim\mathcal{P}_{x,y}} \left[\ell(y, f(x))\right].$$

In practice, sometimes it is impractical to access to $L(f)$. Therefore, an alternative approach is to consider $f$ within a hypothesis class $\mathcal{H}$ instead, and calculate the estimation of population risk based on $\mathcal{S}$, namely the *empirical risk*, to evaluate $f$:

$$\hat{L}(f) = L_n(f; \mathcal{S}) = \frac{1}{n} \sum_{i=1}^{n} \ell(y_i, f(x_i)).$$

Empirical Risk Minimization (ERM) refers to the process we train the model by minimizing the empirical risk defined above. Namely, ERM aims to find an $\hat{f}$ such that

$$\hat{f} = \arg\min_{f \in \mathcal{H}} L_n(f; \mathcal{S}).$$

Note that as in the following part, we mainly consider hypothesis class where function $f$ is fixed, in the remainder of this paper we will re-write it as $L_n(\boldsymbol{\theta}; \mathcal{S})$ to stress the importance of $\boldsymbol{\theta}$ to the empirical risk.

**Mixup training objective.** When applying Mixup (Zhang et al., 2017) to train the model, training samples are used to constructed the mixed samples, and the mixed samples are used to train the model. Following Zhang et al. (2020), we define the mixup training objective as follows:

$$L_n^{\mix}(\boldsymbol{\theta}, \mathcal{S}) = \frac{1}{n^2} \sum_{i,j=1}^{n} \mathbb{E}_{\lambda \sim \mathcal{D}_\lambda} \ell(\boldsymbol{\theta}, \check{z}_{ij}(\lambda)) \tag{1}$$

where $\mathcal{D}_\lambda$ is generally a $Beta(\alpha, \beta)$ distribution with $\alpha = \beta > 0$.

Zhang et al. (2020) proves the relationship between mixup objective and empirical risk minimization:

**Theorem 1.** (Results from Zhang et al. (2020)) *Consider the loss function* $\ell_{\boldsymbol{x}_i, y_i}(\theta) = h(f_\theta(\boldsymbol{x}_i)) - y_i f_\theta(\boldsymbol{x}_i)$. *Denote the standard empirical risk minimization objective as* $L_n^{std}(\boldsymbol{\theta}, \mathcal{S}) = \frac{1}{n} \sum_{i=1}^{n} \ell_{x_i, y_i}(\boldsymbol{\theta})$, *denote* $\tilde{\mathcal{D}}_\lambda = \frac{\alpha}{\alpha+\beta} Beta(\alpha+1, \beta) + \frac{\beta}{\alpha+\beta} Beta(\beta+1, \alpha)$ *the mixture distribution of* $\lambda$, *denote the dataset* $\check{\mathcal{S}} = \{(\check{\boldsymbol{x}}_i, y_i)\}_{i=1}^{n}$, *where* $\check{x}_i = \lambda x_i + (1-\lambda)\boldsymbol{r}_x$, $\boldsymbol{r}_x \sim \mathcal{D}_{n,x}$ *is the empirical distribution of* $\boldsymbol{x}$, $y_i$ *is the original labels for the* $i$-*th training samples in* $\mathcal{S}$. *Then,*

$$L_n^{mix}(\boldsymbol{\theta}, \mathcal{S}) = \mathbb{E}_{\lambda \sim \tilde{\mathcal{D}}_\lambda} \mathbb{E}_{\boldsymbol{r}_x \sim \mathcal{D}_{n,x}} L_n^{std}(\boldsymbol{\theta}, \check{\mathcal{S}}). \tag{2}$$

## 3.2 OmniMixup: A Generalized Version of Mixup with MPSD

In this subection, we propose OmniMixup and show that the recent related works (Zhang et al., 2017; Yao et al., 2022; Baena et al., 2022) can reduce to special cases under OmniMixup. This allow us to analyze all these methods within the same theoretical framework.

**OmniMixup.** For each sample $z_i = (\boldsymbol{x}_i, y_i)$ in the training dataset, OmniMixup defines a Mixing-Pair Sampling Distribution (MPSD) with parameter $z_i$ across the training dataset, i.e., $\boldsymbol{r}_{z_i} = (\boldsymbol{r}_{\boldsymbol{x}_i}, \boldsymbol{r}_{y_i}) \sim \psi_n(z_i), z_i \in \mathcal{S}$. Here $\boldsymbol{r}_{z_i}$ is a random variable with support set $\mathcal{S}$. For simplicity in the theoretical part, we will directly write $\boldsymbol{r}_{z_i} \sim \psi_n(z_i)$. Then, for each sample $z_i$, OmniMixup draws another sample in $\mathcal{S}$ based on $\psi_n(z_i)$ to construct the mixed samples, which are used in the following mixup training. Though the definition is concise, OmniMixup is generalized enough to include many previous related works. We elaborate the empirical risk minimization, vanilla Mixup and C-Mixup, Local-Mixup and Smooth Local-Mixup as follows:

**Example 1** (ERM). *ERM is equivalent to OmniMixup with* $\psi_n(z_j; z_i) = \mathbb{1}(z_j = z_i)$ *(i.e. Dirac delta distribution) given a training sample* $z_i \in \mathcal{S}$, *namely* $z_i$ *will only sample* $z_i$ *itself.*

**Example 2** (vanilla Mixup (Zhang et al., 2017)). *The vanilla Mixup is equivalent to OmniMixup with* $\psi_n(z_j; z_i) = 1/|\mathcal{S}|$ *(i.e. Uniform distribution) given a training sample* $z_i \in \mathcal{S}$, *namely* $z_i$ *will sample data equally.*

**Example 3** (C-Mixup (Yao et al., 2022)). *C-Mixup is equivalent to OmniMixup with* $\psi_n(z_j; z_i) \propto \exp\left(d(y_j, y_i)/2\sigma^2\right)$ *given a training sample* $z_i \in \mathcal{S}$. *Here* $d(.)$ *is a pre-defined distance measure for labels.*

**Example 4** (Local-Mixup (Baena et al., 2022)). *Local-Mixup is equivalent to OmniMixup with MPSD* $\psi_n(z_j; z_i) = \mathbb{1}(d(z_i, z_j)) \leq \epsilon$, *where* $\epsilon$ *is a cut-off value. Here* $d(.)$ *is a pre-defined distance measure for two samples. A smooth version of Local-Mixup is equivalent to OmniMixup with MPSD* $\psi_n(z_j; z_i) \propto \exp(-\alpha \times d(z_i, z_j))$.

It's worth noting that the introduction of MPSDs greatly enhances the flexibility of Mixup. This is because MPSD is not restricted to the similarity-based distribution, which is the previous common practice, MPSDs from random generation, domain expert prior knowledge, or optimization can all be included in the generalized form of OmniMixup.

## 3.3 Generalization Bound of OmniMixup

To theoretically understand MPSD-based mixup strategies, in this subsection, this subsection provides a theoretical analysis of the generalization bound given by OmniMixup. Here we define $\psi(z_i)$

as a MPSD whose support set is equivalent to $\mathcal{P}_{\boldsymbol{x},y}$, and $\psi_n(z_i)$ defined above is the empirical distribution of $\psi(z_i)$.

Inspired by Zhang et al. (2020), this paper proposes to consider the mixup objective as ERM with a regularization term and analyze its second-order Taylor expansion to analyze the generalization bound of OmniMixup. The proof sketch of the analysis can be concluded in four steps: **Step 1**: we connect the OmniMixup training objective with the empirical risk minimization objective; **Step 2**: based on the results of first step, we further obtain the second-order approximation of the regularization term between OmniMixup training objective and the ERM objective under Generalized Linear Model (GLM); **Step 3**: the empirical Rademancher complexity is calculated assuming the model is fitted well; **Step 4**: the generalization bound is directly derived based on the empirical Radmancher complexity according to Bartlett & Mendelson (2002)'s result. Detailed proofs of all theoretical analysis are shown in Appendix A.

**Closed-form of OmniMixup training objective.** To begin with, we first investigate the closed-form of the training objective of the proposed OmniMixup. Specifically, we extend the Eq. 2, which shows the relationship between the vanilla Mixup objective and the ERM objective, to the proposed OmniMixup.

**Corollary 3.1.** *Under OmniMixup, the relationship between the mixup objective and empirical risk minimization is:*

$$L_n^{mix}(\boldsymbol{\theta}, \mathcal{S}) = \mathbb{E}_{\lambda \sim \tilde{\mathcal{D}}_\lambda} \left[ \frac{1}{n} \sum_{i=1}^n \mathbb{E}_{\boldsymbol{r}_{x_i} \sim \psi_n(z_i)} \left[ \ell_{\tilde{x}_i, y_i}(\boldsymbol{\theta}) \right] \right] \tag{3}$$

**Generalized Linear Model.** To analyze the generalization bound, in this section we consider a Generalized Linear Model (GLM), where the model is $f(\boldsymbol{\theta}; x_i) = \boldsymbol{\theta}^\top \boldsymbol{x}_i$, and the empricial training objective is defined as

$$L_n^{std}(\boldsymbol{\theta}; \mathcal{S}) = \frac{1}{n} \sum_{i=1}^n A(\boldsymbol{\theta}^\top \boldsymbol{x}_i) - y_i \boldsymbol{\theta}^T \boldsymbol{x}_i.$$

Here, $A(.)$ is a log-partition function.

Besides, the following assumptions is considered for proving the final result:

**Assumption 1.** $\boldsymbol{\theta}, \mathcal{X}$, *and* $\mathcal{Y}$ *are all bounded.*

**Assumption 2.** *The expectation of* $\boldsymbol{r}_{x_i} \sim \psi(\boldsymbol{z}_i)$ *is* $\boldsymbol{0}$.

**OmniMixup as a regularization term** A common practice to consider the relationship between mixup objective and ERM objective is to view the former one as a ERM objective with a regularization term (Zhang et al., 2020; Park et al., 2022a). Similarly, in this paper, we connect the training objective of OmniMixup to ERM objective with a regularization term via Lemma 3.1.

**Lemma 3.1.** *Denote* $\hat{\Sigma}_{\boldsymbol{x}_i}$ *as the estimate of variance of* $\psi(\boldsymbol{x}_i)$. *For a GLM, if* $A(\cdot)$ *is twice differentiable, then*

$$L_n^{mix}(\boldsymbol{\theta}, \mathcal{S}) = L_n^{std}(\boldsymbol{\theta}, \mathcal{S}) + \frac{\mathbb{E}_{\lambda \sim \tilde{\mathcal{D}}}(1-\lambda)^2}{2n\bar{\lambda}^2} \sum_{i=1}^n \left[ A''(\boldsymbol{x}_i^\top \boldsymbol{\theta}) \cdot \boldsymbol{\theta}^\top \hat{\Sigma}_{\psi(z_i)} \boldsymbol{\theta} \right] \tag{4}$$

**Generalization bound** To analyze the generalization bound of the OmniMixup objective, we adopt an common approach to first analyze the empirical Radmancher complexity of a given hypothesis class and the training dataset. Specifically, we first make assumptions about the hypothesis class considered and the sampling distributions.

**Assumption 3.** *Denote* $\Sigma_{\psi(z_i)}$ *as the variance of* $\boldsymbol{r}_{\boldsymbol{x}_i} \sim \psi(z_i)$, *the following hypothesis class is considered when analyze the Radmancher complexity:*

$$\mathcal{W}_\gamma := \{ \boldsymbol{\theta} \mid \forall\, i \in [n], \mathbb{E}_{\boldsymbol{x} \sim \psi(z_i)} A''(\boldsymbol{\theta}^\top \boldsymbol{x}) \cdot \boldsymbol{\theta}^\top \Sigma_{\psi(z_i)} \boldsymbol{\theta} \le \gamma \},$$

Note that such assumption of the hypothesis class is reasonable, as it considers parameters space where the regularization term proved in Lemma 3.1 is minimized properly, suggesting that the OmniMixup strategy works well during the optimization process.

**Assumption 4.** $\forall\, i \in [n]$, $\psi(z_i)$ *is $\rho$-retentive, $\rho \in (0, 1/2]$.*

The definition of $\rho$-retentive is defined as below:

**Definition 1.** *A probability distribution $p(\boldsymbol{x})$ is $\rho$-retentive if for any non-zero vector $\boldsymbol{v} \in \mathbb{R}^d$,*

$$\left[\mathbb{E}_{\boldsymbol{x}}[A''(\boldsymbol{x}^\top \boldsymbol{v})]\right]^2 \geq \rho \cdot \min\{1, \mathbb{E}_{\boldsymbol{x}}(\boldsymbol{x}^\top \boldsymbol{v})^2\}.$$

Accordingly, we have the following lemma providing an upper bound for the empirical Radmancher complexity.

**Lemma 3.2.** *The Rademacher complexity of $\mathcal{W}_\gamma$ satisfies*

$$Rad(\mathcal{W}_\gamma, \mathcal{S}) \leq \sqrt{\frac{\eta}{n} \mathbb{E}_{z \sim \mathcal{P}_{x,y}}[\boldsymbol{x}^\top \Sigma_{\psi(z)}^{-1} \boldsymbol{x}]},$$

*where $\eta = \max\{(\frac{\gamma}{\rho})^{1/2}, (\frac{\gamma}{\rho})\}$, $\Sigma_{\psi(z)}$ is the covariance matrix of distribution $\psi(z)$.*

Based on this bound on Rademacher complexity, we can directly obtain the generalization bound.

**Theorem 2.** *Assume $A(\cdot)$ be $L$-Lipschitz continuous, $\mathcal{X}$, $\mathcal{Y}$ and $\boldsymbol{\theta}$ are bounded, then there exists constants $L, B > 0$, such that $\forall\, \boldsymbol{\theta} \in \mathcal{W}_\gamma$, which is the regularization induced by Mixup, we have*

$$L(\boldsymbol{\theta}) \leq L_n^{std}(\boldsymbol{\theta}, \mathcal{S}) + 2L \cdot L_A \cdot \sqrt{\frac{\eta}{n} \mathbb{E}_{z \sim \mathcal{P}_{x,y}}[\boldsymbol{x}^\top \Sigma_{\psi(z)}^{-1} \boldsymbol{x}]} + B\sqrt{\frac{\log(1/\delta)}{2n}},$$

*with probability at least $1 - \delta$.*

### 3.4 OMNIEVAL: AN EVALUATION FRAMEWORK FOR MPSDS WITHIN OMNIMIXUP

In this subsection, we explain the theoretical result in Theorem 2, and provide an insight of comparison among different MPSDs for Om-niMixup. Based on this insight, we propose OmniEval, an evaluation framework that is able to measure the effectiveness of a given MPSD, and automatically search for the best MPSD. Note that the comparison between the vanilla Mixup and the ERM has been discussed in previous work [1], the comparison of OmniMixup and ERM is therefore beyond the scope of this work, as we can compare them indirectly via the vanilla Mixup, which is also a special case of OmniMixup.

From Theorem 2, it is clear that the upper bound of the generalization gap is strongly related to $\mathbb{E}_{z \sim \mathcal{P}_{x,y}}[\boldsymbol{x}^\top \Sigma_{\psi(z)}^{-1} \boldsymbol{x}]$. In the remainder of the paper, we will call this quantity *expected M-Score*, as the quantity inside the expectation is Mahalanobis distance, which is used to measure the distance between samples and distribution. This suggests that given a training dataset $\mathcal{S}$, comparing the generalization ability of different OmniMixup strategies can be reduced to comparing only the expected M-Score among different MPSDs.

However, two challenges remain in order to

---

**Algorithm 1** OmniEval

**Input:** A set of MPSDs $\Psi_n$, a training dataset $\mathcal{S}$, a model $f_{\boldsymbol{\theta}}$ with parameters $\boldsymbol{\theta}$.

**Output:** An MPSD $\psi_n$.

1: Step 1: Train a model under ERM and obtain $\boldsymbol{X}$.
2: Train $f$ over $\mathcal{S}$ with ERM and obtain parameters $\boldsymbol{\theta}^*$.
3: $\boldsymbol{X} = \{\}$
4: **for** $z \in \mathcal{S}$ **do**
5:     $\boldsymbol{X} = \boldsymbol{X} \cup \{\boldsymbol{x}_z\}$, $\boldsymbol{x}_z$ is the encoded features of $z$ before final linear layer in $f_{\boldsymbol{\theta}^*}$.
6: **end for**
7: Step 2: Calculate M-Score estimate for each MPSD.
8: $\hat{\mathcal{M}}_{\psi_n}^* = \infty$.
9: $\psi_n^* = \texttt{NONE}$.
10: **for** $\psi_n \in \Psi_n$ **do**
11:     Access to probability matrix $\boldsymbol{A} \in \mathbb{R}^{|\mathcal{S}| \times |\mathcal{S}|}$ of $\psi_n$.
12:     $\Sigma = \texttt{weightedCov}(\boldsymbol{X}, \boldsymbol{A})$
13:     $\hat{\mathcal{M}}_{\psi_n} = \{\}$.
14:     **for** $\boldsymbol{x}_i \in \boldsymbol{X}$ **do**
15:         $\hat{\mathcal{M}}_{\psi_n} = \hat{\mathcal{M}}_{\psi_n} \cup \{\boldsymbol{x}_i^\top \Sigma^{-1} \boldsymbol{x}_i\}$.
16:     **end for**
17:     $\psi_n^* = \psi_n$ if $\hat{\mathcal{M}}_{\psi_n} = \texttt{min}(\hat{\mathcal{M}}_{\psi_n}^*, \hat{\mathcal{M}}_{\psi_n})$.
18:     $\hat{\mathcal{M}}_{\psi_n}^* = \texttt{min}(\hat{\mathcal{M}}_{\psi_n}^*, \hat{\mathcal{M}}_{\psi_n})$.
19: **end for**
20: Step 3: Return the best MPSD.
21: **return** $\psi_n^*$

---

compare M-Score of different MPSDs. First, as latent features in real applications, $\boldsymbol{x}$ is inaccessible unless the model is trained. This make comparison expensive as we have to train models with

---

[1] Zhang et al. (2020) showed that the vanilla Mixup approach has tighter generalization upper bound if the intrinsic dimension of $x$ is small.

all MPSDs one by one to make comparisons. While once the model is trained, there is no need to compare the expected M-Score anymore; second, the calculation $\mathbb{E}_{z \sim \mathcal{P}_{x,y}}[\boldsymbol{x}^\top \Sigma_{\psi(z)}^{-1} \boldsymbol{x}]$ is intractable. To address these challenges, we propose OmniEval to automatically search for MPSDs that has great potential effectiveness. Specifically, we propose to train a model with training dataset $\mathcal{S}$ first under ERM fashion and save the features before the final linear layer of all data in $\mathcal{S}$. Then, we use the saved features to give an Method of Moment (MoM) estimator $\hat{\mathcal{M}} = \frac{1}{n} \sum_{i=1}^n \boldsymbol{x}_i^\top \Sigma_{\psi(z_i)}^{-1} \boldsymbol{x}_i$.

We return the MPSD mixup with the smallest $\hat{\mathcal{M}}$ to use for training the model. The algorithm of OmniEval is summarized in Algorithm 1.

## 4 IMPLEMENTATION DETAILS

In this section, we present the implementation details of OmniMixup in our empirical study.

### 4.1 SETTINGS OF MPSDS

In this subsection, we present the implementation details of MPSD for OmniMixup in data in two different domains. As the search space of MPSD is huge, it is impractical to search over all the possible MPSD and calculate its corresponding M-Score. Therefore, we restrict the search space into several specific sets of MPSDs that we use to search for the best MPSD and train the model.

**Image Classification**  For image classification tasks, inspired by the smooth LocalMixup Baena et al. (2022), given a batch of images $\mathcal{B} = \{I_1, ..., I_b\}$, we apply the current popular vision-language model CLIP to obtain the representation of images, denoted as $\boldsymbol{h} = \{h_1, ..., h_b\}$, and design a family of MPSD as follows:

$$\Psi = \{\psi_n^{\tau,\beta} | \tau \in T; \beta \in B\}, \text{ where } \psi_n^{\tau,\beta}(I_i) = \texttt{softmax}\left(\tau \times \exp\left(-\beta \times dis(h_i, \boldsymbol{h})\right)\right) \in \mathbb{R}^b.$$

Here $\tau$ and $\beta$ are both hyperparameters selected from sets $T$ and $B$, respectively.

**Molecular Property Prediction**  For molecular property prediction, we restrict MPSD into three similarity-based families based on either molecular fingerprints and training labels: 1) fingerprint-MPSD (fp-MPSD); 2) inverse-fingerprint-MPSD (invfp-MPSD); 3) labels, namely C-Mixup. Specifically, for fp/invfp-MPSD, given a batch of $d$-dimensional fingerprints $\boldsymbol{m} = \{\boldsymbol{m}_1, ..., \boldsymbol{m}_b\} \in \{0,1\}^{d \times b}$ of molecules $\{M_1, ..., M_b\}$ in a mini-batch, the sampling distribution of $M_i$ is defined as follows:

$$\psi_{n,fp}(M_i) = \texttt{softmax}\left(\tau \times \exp\left(-\beta \frac{\texttt{Manhattan}(\boldsymbol{m}_i, \boldsymbol{m})}{d}\right)\right) \in \mathbb{R}^b.$$

$$\psi_{n,invfp}(M_i) = \texttt{softmax}\left(\tau \times \exp\left(-\beta \frac{(d - \texttt{Manhattan}(\boldsymbol{m}_i, \boldsymbol{m}))}{d}\right)\right) \in \mathbb{R}^b.$$

In terms of label-MPSD, suppose $y_i \in R^n, i = 1, ..., b$ are labels of input data $M_i$, then the design of MPSD is motivated by C-Mixup (Yao et al., 2022):

$$\psi_{n,labels}(M_i) = \texttt{softmax}\left(\tau \times \exp\left(-\beta \frac{\texttt{dis}(y_i, y)}{d}\right)\right) \in \mathbb{R}^b.$$

Following C-Mixup (Yao et al., 2022), we restrict the use of this family for regression tasks only. In our experiments, we aim to identify the optimal MPSD from the three families. Specifically, we employ a grid search for $\tau$ and $\beta$ to search for the best combination leading to MPSD with minimal M-Score. The resulting MPSD is then utilized in training. Although the current search method prioritizes efficiency over the full exploration of the probability space, it is worth noting that an optimization-based search might produce a more refined MPSD, which we intend to explore in future work.

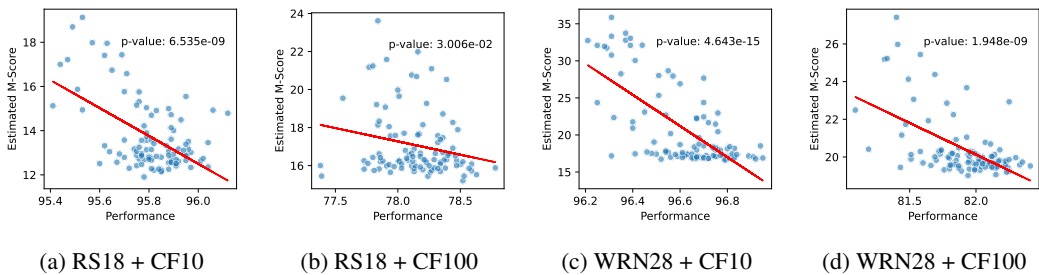

(a) RS18 + CF10          (b) RS18 + CF100          (c) WRN28 + CF10          (d) WRN28 + CF100

Figure 1: Relationship between the estimated M-Score and their respective model performances. The p-value represents the significance level of the association between M-Score and model performance.

## 4.2 BASELINE

For both domains, we compare the OmniMixup with ERM and the vanilla Mixup under the same backbone models. In terms of the backbone models, we select PreActResNet18 (He et al., 2016), WideResNet28-19 (Zagoruyko & Komodakis, 2016), and DenseNet190 (Huang et al., 2017) for our experiments. For molecular property prediction, we take Uni-Mol (Zhou et al., 2023) as the backbone model. The baselines presented in this paper are all re-implemented with recommended hyperparameter settings from the original papers.

## 5 EXPERIMENTS

In this section, we empirically investigate whether the framework proposed in this paper can resolve challenges mentioned in § 1. Specifically, we want to answer the following research questions:

- **RQ1:** Can we use the estimated M-Score presented in theoretical analysis to obtain insight about the potential effectiveness of designed MPSD? This research question is used to verify the effectiveness of OmniEval;

- **RQ2:** Can OmniMixup selected from OmniEval be easily applied across different modalities and domains? This research question is used to verify whether OmniMixup can be applied to diverse situations without prior knowledge.

## 5.1 A1: ESTIMATED M-SCORE PROVIDE POTENTIAL INSIGHT TOWARD THE EFFECTIVENESS OF MPSD

To answer the first research question, we conduct experiments to investigate the relationship between the M-Score of MPSD and the corresponding performance. Specifically, we train ResNet18 and WideResNet28-10 over CIFAR-10 and CIFAR100. Specifically, we apply the MPSDs family presented in § 4.1 with fixed $\tau$ and $\beta$ sampled from $[0, 1]$ to investigate the relationship between M-Score and the effectiveness of the model. The results are shown in Figure 1.

From the result, it is clear to find out that: 1) the estimated M-Score of MPSD is significantly negatively associated with the performance of models trained under the corresponding mixup strategies. Specifically, a higher M-Score is generally associated with poorer performance; 2) a weaker association appears in Figure 1, suggesting that the informative values may also be restricted by the poor performance capacity of the model. This finding is actually aligned with the Assumption 3, which assumes that the models should fit well on the mixed virtual data.

## 5.2 A2: THE PROPOSED OMNIMIXUP CAN BE EASILY APPLIED TO DIFFERENT MODALITIES AND DOMAINS.

To verify whether our proposed method can be easily applied to various modalities and domains, we apply OmniMixup with OmniEval pipeline on image classification benchmarks mentioned above and eight molecular property prediction tasks selected from MoleculeNet (Wu et al., 2018). The experimental results are shown in Table 1 and Table 2.

Table 1: Overall performance of OmniMixup on image classification benchmarks.

| | ResNet18 | | WideResNet28-10 | | DenseNet190 | |
| | CIFAR-10 | CIFAR-100 | CIFAR-10 | CIFAR-100 | CIFAR-10 | CIFAR-100 |
|---|---|---|---|---|---|---|
| ERM | 94.4 | 75.7 | 95.5 | 78.9 | 95.9 | 80.9 |
| Mixup | 95.8 | 78.6 | 96.7 | 81.9 | 97.2 | **83.9** |
| OmniMixup | **96.1** | **78.8** | **96.9** | **82.5** | **97.5** | 83.7 |

Table 2: Overall performance of Mixup approaches on the molecular property prediction benchmark. All experiments are mean of 3 runs. Numbers within parentheses are standard deviations of the performances.

| | Classification (Higher is better) | | | | Regression (Lower is better) | | | |
| Dataset | BACE | BBBP | ClinTox | SIDER | ESOL | FreeSolv | Lipo | QM7 |
|---|---|---|---|---|---|---|---|---|
| Uni-Mol | 0.862 (0.004) | 0.737 (0.005) | 0.932 (0.004) | 0.658 (0.020) | 0.812 (0.016) | 1.605 (0.058) | 0.606 (0.003) | 42.94 (0.158) |
| Mixup | 0.876 (0.008) | 0.740 (0.003) | 0.899 (0.012) | 0.662 (0.002) | 0.796 (0.010) | **1.571** (**0.074**) | 0.590 (0.003) | 43.43 (1.124) |
| OmniMixup | **0.886** (**0.011**) | **0.742** (**0.008**) | **0.949** (**0.013**) | **0.673** (**0.004**) | **0.795** (**0.023**) | 1.574 (0.102) | **0.589** (**0.015**) | **41.41** (**2.064**) |

Table 3: Values of the estimated M-Score.

| Dataset | BACE | BBBP | ClinTox | SIDER | ESOL | FreeSolv | Lipo | QM7 |
|---|---|---|---|---|---|---|---|---|
| Mixup | 31.126 | 30.012 | 36.835 | 31.898 | 40.354 | 694.761 | 25.546 | 28.721 |
| OmniMixup | 31.059 | 29.949 | 35.221 | 31.850 | 40.280 | 645.604 | 25.450 | 28.553 |

From Table 1, the OmniMixup consistently outperforms the vanilla Mixup and the ERM baseline across almost all different datasets and different models. In terms of the molecular property prediction benchmark in Table 2, the performance of OmniMixup surpasses the Uni-Mol baseline and the vanilla Mixup approach. Both results demonstrate that 1) the proposed OmniEval pipeline can efficiently search MPSD based on M-Score and gain improvement for the model's performance; 2) the OmniEval pipeline is suitable for general use across modalities and domains.

### 5.3 ANALYSIS OF M-SCORE

Additionally, in this subsection, we provide the estimates of the M-Score of the vanilla mixup and the M-Score of the best MPSD mixup we identified in Table 3. We find that: 1) within the defined families, although search space is limited, **we consistently identified mixups with an M-Score lower than the vanilla mixup, which indicates that the vanilla mixup is still far from the optimal one.** We look forward to exploring more powerful search methods in future work; 2) **there appears to be a relationship between the M-Score difference and the final performance of the model.** There is a large difference between the M-Score of vanilla mixup and MPSD mixup in the Clintox dataset, so as the final model performance; 3) finally, we find that while the gap in M-Score for SIDER is insignificant, the model effect gap amounts to 1.1%. In contrast, though we gain a large improvement on M-Score in FreeSolv, the resulting performance is conversely bad, compared to the vanilla mixup. We consider this inaccurate information can also originate from Assumption 3, which we have elaborated in § 5.1.

## 6 CONCLUSION

We proposes OmniMixup, a versatile mixup technique that is applicable across modalities and domains. OmniMixup generalizes the vanilla Mixup (Zhang et al., 2017) and thereby includes previous related works into a holistic framework. A theoretical analysis is further conducted based on the unified framework to investigate the generalization ability of the OmniMixup. Based on the theoretical result, an evaluation pipeline OmniEval based on M-Score is developed to identify the optimal MPSD for OmniMixup. The empirical study shows that: 1) M-Score in OmniEval is insightful about the generalization ability of OmniMixup; 2) along with OmniEval, OmniMixup can provide improvement in performance for models regardless of modalities of data and domains of tasks.

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

# A PROOFS

## A.1 PROOF OF COROLLARY 3.1

*Proof.*

$$L_n^{\text{mix}}(\boldsymbol{\theta}, \mathcal{S}) = \mathbb{E}_{\lambda \sim \tilde{\mathcal{D}}_\lambda} \mathbb{E}_{\boldsymbol{r}_x \sim \mathcal{D}_{n,x}} L_n^{\text{std}}(\boldsymbol{\theta}, \check{\mathcal{S}})$$

$$= \mathbb{E}_{\lambda \sim \tilde{\mathcal{D}}_\lambda} \mathbb{E}_{\boldsymbol{r}_x \sim \mathcal{D}_{n,x}} \frac{1}{n} \sum_{i=1}^{n} \ell_{\check{x}_i, y_i}(\boldsymbol{\theta})$$

$$= \mathbb{E}_{\lambda \sim \tilde{\mathcal{D}}_\lambda} \left[ \frac{1}{n} \sum_{i=1}^{n} \mathbb{E}_{\boldsymbol{r}_x \sim \mathcal{D}_{n,x}} [\ell_{\check{x}_i, y_i}(\boldsymbol{\theta})] \right].$$

The result can be proved by substituting $x \sim \mathcal{D}_{n,x}$ with data-specific random variables $\boldsymbol{r}_{x_i} \sim \psi_n(x_i)$ in $\check{\boldsymbol{x}}_i$ and take expectation correspondingly.

$\square$

## A.2 PROOF OF LEMMA 3.1

Note that Lemma A.1 and Lemma A.2 are needed for proving the result.

*Proof.* As GLM is invariant of scaling, here we use a normalized mixup training dataset $\tilde{S} = \{(\tilde{\boldsymbol{x}}_i, y_i)\}_{i=1}^{n}$ with $\tilde{\boldsymbol{x}}_i = \frac{1}{\lambda}(\lambda \boldsymbol{x}_i + (1 - \lambda)\boldsymbol{r}_{\boldsymbol{x}_i})$ accordingly to simplify the proof.

According to Corollary 3.1,

$$L_n^{\text{mix}}(\boldsymbol{\theta}, \mathcal{S}) - L_n^{\text{std}}(\boldsymbol{\theta}, \mathcal{S}) = \mathbb{E}_{\lambda \sim \tilde{\mathcal{D}}_\lambda} \left[ \frac{1}{n} \sum_{i=1}^{n} \mathbb{E}_{\boldsymbol{r}_{\boldsymbol{x}_i} \sim \psi_n(\boldsymbol{x}_i)} [\ell_{\tilde{\boldsymbol{x}}_i, y_i}(\boldsymbol{\theta})] \right] - \frac{1}{n} \sum_{i=1}^{n} \ell_{\boldsymbol{x}_i, y_i}(\boldsymbol{\theta})$$

$$= \mathbb{E}_{\lambda \sim \tilde{\mathcal{D}}_\lambda} \left[ \frac{1}{n} \sum_{i=1}^{n} \mathbb{E}_{\boldsymbol{r}_{\boldsymbol{x}_i} \sim \psi_n(\boldsymbol{x}_i)} [\ell_{\tilde{\boldsymbol{x}}_i, y_i}(\boldsymbol{\theta}) - \ell_{\boldsymbol{x}_i, y_i}(\boldsymbol{\theta})] \right]$$

$$= \mathbb{E}_{\xi} \left[ \frac{1}{n} \sum_{i=1}^{n} [\ell_{\tilde{\boldsymbol{x}}_i, y_i}(\boldsymbol{\theta}) - \ell_{\boldsymbol{x}_i, y_i}(\boldsymbol{\theta})] \right]$$

Here $\xi = (\lambda, \boldsymbol{r}_{x_1}, ..., \boldsymbol{r}_{x_n})$ just for simplicity. From above we know that

$$\frac{1}{n} \sum_{i=1}^{n} \ell_{\tilde{\boldsymbol{x}}_i, y_i}(\boldsymbol{\theta}) - \ell_{\boldsymbol{x}_i, y_i}(\boldsymbol{\theta}) = \frac{1}{n} \sum_{i=1}^{n} -\left(y_i \tilde{\boldsymbol{x}}_i^\top \boldsymbol{\theta} - A(\tilde{\boldsymbol{x}}_i^\top \boldsymbol{\theta})\right) - \frac{1}{n} \sum_{i=1}^{n} -\left(y_i \boldsymbol{x}_i^\top \boldsymbol{\theta} - A(\boldsymbol{x}_i^\top \boldsymbol{\theta})\right)$$

$$= \frac{1}{n} \sum_{i=1}^{n} (y_i \boldsymbol{x}_i^\top \boldsymbol{\theta} - y_i \tilde{\boldsymbol{x}}_i^\top \boldsymbol{\theta}) + \frac{1}{n} \sum_{i=1}^{n} (A(\tilde{\boldsymbol{x}}_i^\top \boldsymbol{\theta}) - A(\boldsymbol{x}_i^\top \boldsymbol{\theta}))$$

We can prove that $\mathbb{E}_{\xi} \left[ \frac{1}{n} \sum_{i=1}^{n} (y_i \boldsymbol{x}_i^\top \boldsymbol{\theta} - y_i \tilde{\boldsymbol{x}}_i^\top \boldsymbol{\theta}) \right] = 0$. For each $i \in [n]$ of second term, taking taylor expansion on $\tilde{\boldsymbol{x}}_i = \boldsymbol{x}_i$, we have

$$A(\tilde{\boldsymbol{x}}_i^\top \boldsymbol{\theta}) - A(\boldsymbol{x}_i^\top \boldsymbol{\theta}) \approx A'(\boldsymbol{x}_i^\top \boldsymbol{\theta})(\tilde{\boldsymbol{x}}_i - \boldsymbol{x}_i)^\top \boldsymbol{\theta} + \frac{1}{2} A''(\boldsymbol{x}_i^\top \boldsymbol{\theta}) \boldsymbol{\theta}^\top (\tilde{\boldsymbol{x}}_i - \boldsymbol{x}_i)(\tilde{\boldsymbol{x}}_i - \boldsymbol{x}_i)^\top \boldsymbol{\theta} \quad (5)$$

Taking expectation over Eq. (5), we have

$$\mathbb{E}_{\xi}[A(\tilde{\boldsymbol{x}}_i^\top \boldsymbol{\theta}) - A(\boldsymbol{x}_i^\top \boldsymbol{\theta})] = \frac{\mathbb{E}_{\lambda \sim \tilde{\mathcal{D}}}(1 - \lambda)^2}{2\bar{\lambda}^2} A''(\boldsymbol{x}_i^\top \boldsymbol{\theta}) \boldsymbol{\theta}^\top \hat{\Sigma}_{x_i} \boldsymbol{\theta}.$$

Plugging in back to above proves the result. $\square$

**Lemma A.1.** $\mathbb{E}_\xi[\tilde{\boldsymbol{x}}_i - \boldsymbol{x}_i] = 0$.

*Proof.* Based on the assumption of $\mathcal{D}_{\boldsymbol{x}_i}$, we have

$$
\begin{aligned}
\mathbb{E}_\xi[\tilde{\boldsymbol{x}}_i] &= \mathbb{E}_\xi\left[\frac{\lambda\boldsymbol{x}_i + (1-\lambda)\boldsymbol{r}_{\boldsymbol{x}_i}}{\bar{\lambda}}\right] \\
&= \frac{\mathbb{E}_\xi[\lambda]\boldsymbol{x}_i + (1 - \mathbb{E}_\xi[\lambda])\mathbb{E}_\xi[\boldsymbol{r}_{\boldsymbol{x}_i}]}{\bar{\lambda}} \\
&= \boldsymbol{x}_i.
\end{aligned}
$$

$\square$

**Lemma A.2.** $\mathbb{E}_\xi[(\tilde{\boldsymbol{x}}_i - \boldsymbol{x}_i)(\tilde{\boldsymbol{x}}_i - \boldsymbol{x}_i)^\top] = \frac{\mathbb{E}_\lambda(1-\lambda)^2}{\bar{\lambda}}\hat{\Sigma}_{\boldsymbol{x}_i}$, *where* $\hat{\Sigma}_{\boldsymbol{x}_i} := Var(\boldsymbol{r}_{\boldsymbol{x}_i}) = \mathbb{E}[\boldsymbol{r}_{\boldsymbol{x}_i}\boldsymbol{r}_{\boldsymbol{x}_i}^\top]$.

*Proof.* Accordingly, we know that the LHS equals to

$$
\begin{aligned}
LHS &= \mathbb{E}_\xi\left[\tilde{\boldsymbol{x}}_i\tilde{\boldsymbol{x}}_i^\top - \boldsymbol{x}_i\tilde{\boldsymbol{x}}_i^\top - \boldsymbol{x}_i\tilde{\boldsymbol{x}}_i^\top + \boldsymbol{x}_i\boldsymbol{x}_i^\top\right] \\
&= \mathbb{E}_\xi\left[\frac{1}{\bar{\lambda}^2}(\lambda\boldsymbol{x}_i + (1-\lambda)\boldsymbol{r}_{\boldsymbol{x}_i})(\lambda x_i + (1-\lambda)\boldsymbol{r}_{\boldsymbol{x}_i})^\top\right] - \boldsymbol{x}_i\boldsymbol{x}_i^\top \\
&= \mathbb{E}_\xi\left[\frac{1}{\bar{\lambda}^2}\lambda^2\boldsymbol{x}_i\boldsymbol{x}_i^\top + \frac{(1-\lambda)^2}{\bar{\lambda}^2}\boldsymbol{r}_{\boldsymbol{x}_i}\boldsymbol{r}_{\boldsymbol{x}_i}^\top\right] - \boldsymbol{x}_i\boldsymbol{x}_i^\top \\
&= \frac{\mathbb{E}_\lambda[(1-\lambda)^2]}{\bar{\lambda}^2}\mathbb{E}_{\boldsymbol{r}_{\boldsymbol{x}_i}\sim\mathcal{D}_{\boldsymbol{x}_i}}[\boldsymbol{r}_{\boldsymbol{x}_i}\boldsymbol{r}_{\boldsymbol{x}_i}^T] \\
&= \frac{\mathbb{E}_\lambda[(1-\lambda)^2]}{\bar{\lambda}^2}\hat{\Sigma}_{\boldsymbol{x}_i}.
\end{aligned}
$$

$\square$

## A.3 PROOF OF LEMMA A.3

*Proof.* We prove from the definition of empirical Radmancher complexity.

By definition, given $n$ *i.i.d.* Rademacher r.v. $\xi_1, ..., \xi_n$, the empirical Rademacher complexity is

$$
Rad(\mathcal{W}_\gamma, \mathcal{S}) = \mathbb{E}_\xi\left[\sup_{\boldsymbol{\theta}\in\mathcal{W}_\gamma}\frac{1}{n}\sum_{i=1}^n \xi_i\boldsymbol{\theta}^\top\boldsymbol{x}_i\right].
$$

Let $\tilde{\boldsymbol{x}}_i = \Sigma_{\boldsymbol{x}_i}^{-1/2}\boldsymbol{x}_i$, $a_i(\boldsymbol{\theta}) = \mathbb{E}_{\boldsymbol{x}\sim\psi(\boldsymbol{x}_i)}[A''(\boldsymbol{x}^\top\boldsymbol{\theta})]$ and $\boldsymbol{v}_i = \Sigma_{\boldsymbol{x}_i}^{1/2}\boldsymbol{\theta}$, then $\rho$-retentiveness condition implies $a_i(\boldsymbol{\theta})^2 \geq \rho \cdot \min\{1, \mathbb{E}_{\boldsymbol{x}\sim\psi(\boldsymbol{x}_i)}(\boldsymbol{\theta}^\top\boldsymbol{x})^2\} \geq \rho \cdot \min\{1, \boldsymbol{\theta}^\top\Sigma_{\boldsymbol{x}_i}\boldsymbol{\theta}\}$ and therefore $a_i(\boldsymbol{\theta}) \cdot \boldsymbol{\theta}^\top\Sigma_{\boldsymbol{x}_i}\boldsymbol{\theta} \leq \gamma$ implies that $\|\boldsymbol{v}_i\|^2 = \boldsymbol{\theta}^\top\Sigma_{\boldsymbol{x}_i}\boldsymbol{\theta} \leq \max\{(\frac{\gamma}{\rho})^{1/2}, \frac{\gamma}{\rho}\} = \beta$.

Hence,

$$
\begin{aligned}
Rad(\mathcal{W}_\gamma, \mathcal{S}) =& \mathbb{E}_\xi \sup_{\boldsymbol{\theta} \in \mathcal{W}_\gamma} \frac{1}{n} \sum_{i=1}^n \xi_i \boldsymbol{\theta}^\top \boldsymbol{x}_i \\
=& \mathbb{E}_\xi \sup_{\boldsymbol{\theta} \in \mathcal{W}_\gamma} \frac{1}{n} \sum_{i=1}^n \xi_i \boldsymbol{v}_i^\top \tilde{\boldsymbol{x}}_i \\
\leq& \mathbb{E}_\xi \sup_{\|\boldsymbol{v}_i\|^2 \leq \eta} \frac{1}{n} \sum_{i=1}^n \xi_i \boldsymbol{v}_i^\top \tilde{\boldsymbol{x}}_i \\
\leq& \mathbb{E}_\xi \sup_{\|\boldsymbol{v}_i\|^2 \leq \eta} \frac{1}{n} \sum_{i=1}^n \|\boldsymbol{v}_i\| \cdot \|\xi_i \tilde{\boldsymbol{x}}_i\| \qquad \text{(Cauchy-Schwarz Inequality)} \\
\leq& \frac{\sqrt{\eta}}{n} \cdot \mathbb{E}_\xi \| \sum_{i=1}^n \xi_i \tilde{\boldsymbol{x}}_i \| \\
\leq& \frac{\sqrt{\eta}}{n} \cdot \sqrt{\left( \mathbb{E}_\xi \| \sum_{i=1}^n \xi_i \tilde{\boldsymbol{x}}_i \| \right)^2} \\
\leq& \frac{\sqrt{\eta}}{n} \cdot \sqrt{\mathbb{E}_\xi \| \sum_{i=1}^n \xi_i \tilde{\boldsymbol{x}}_i \|^2} \qquad \text{(Jensen's Inequality)} \\
\leq& \frac{\sqrt{\eta}}{n} \cdot \sqrt{\sum_{i=1}^n \tilde{\boldsymbol{x}}_i^\top \tilde{\boldsymbol{x}}_i} . \qquad \text{(Triangle Inequality)}
\end{aligned}
$$

Taking expectation over the whole dataset, we have

$$
\begin{aligned}
Rad(\mathcal{W}_\gamma, \mathcal{S}) = \mathbb{E}_S[Rad(\mathcal{W}_\gamma, \mathcal{S})] \leq& \frac{\sqrt{\eta}}{n} \cdot \sqrt{\sum_{i=1}^n \mathbb{E}_{z \sim \mathcal{P}_{x,y}}[\tilde{\boldsymbol{x}}_i^\top \tilde{\boldsymbol{x}}_i]} \qquad \text{(Jensen's Inequality)} \\
\leq& \sqrt{\frac{\eta}{n} \mathbb{E}_{z \sim \mathcal{P}_{x,y}}[\boldsymbol{x}^\top \Sigma_{\psi(z)}^{-1} \boldsymbol{x}]}.
\end{aligned}
$$

$\square$

### A.4 PROOF OF THEOREM 2

*Proof.* This results is directly proved by applying Lemma A.3.

**Lemma A.3** (Result from Bartlett & Mendelson (2002)). *For any $B$-uniformly bounded and $L$-Lipchitz function $\zeta$, for all $\phi \in \Phi$, with probability at least $1 - \delta$,*

$$
\mathbb{E}\zeta(\phi(x_i)) \leq \frac{1}{n} \sum_{i=1}^n \zeta(\phi(x_i)) + 2L Rad(\Phi, \mathcal{S}) + B\sqrt{\frac{\log(1/\delta)}{2n}}.
$$

$\square$

## B EXPERIMENTAL DETAILS

### B.1 DATASET

In this subsection, we provide details of the datasets used in experiments. For image classification tasks, we utilize the CIFAR-10 and CIFAR-100 datasets, which are frequently utilized in recent Mixup works. For the molecular domain, we conduct experiments on eight tasks in MoleculeNet (Wu et al., 2018) benchmark: BACE, BBBP, ClinTox, SIDER, ESOL, FreeSolve, Lipo, QM7.

A primary reason for selecting these datasets is their limited training data size. For molecular property prediction, we follow Zhou et al. (2023) to split the datasets into train/validation/test splits.

## B.2 HYPERPARAMETER SETTING

In this subsection, we present the hyperparameter settings used for our empirical study.

For the image classification benchmarks, we follow the hyperparameter settings from the vanilla mixup (Zhang et al., 2017) except that we select $\alpha$ from $[0, 2]$. For the molecular property prediction benchmark, we follow the hyperparameter setting of Zhou et al. (2023) to re-implement baselines. We grid search learning rate from $\{0.0003, 0.0001, 8e-05, 5e-05, 3e-05, 2e-05, 1e-05\}$, batch size from $\{8, 16, 32, 64, 128\}$, $\alpha$ from $\{0.1, 0.2, 0.5, 1, 2\}$. All the experiments are run three times and report the mean and standard variance.

