# OpenReview forum: "OmniMixup: Generalize Mixup with Mixing-Pair Sampling Distribution"
_ICLR.cc/2024/Conference — Submitted to ICLR 2024_

### Official Review · Reviewer_sgeX · 2023-10-19

**Soundness:** 2 fair
**Presentation:** 2 fair
**Contribution:** 2 fair
**Rating:** 3
**Confidence:** 5

**Summary:**

This paper studies the general form of sample mixup by proposing OmniMixup that learns the optimal sampling strategies to sample instances. The core idea is to generalize existing sampling selection techniques to a learnable function. Then, authors analyzed the generalization bound of this approach. Experiments on image classification and molecular property prediction demonstrate the effectiveness of this approach.

**Strengths:**

1. The idea of generalizing all sampling strategies into one unified framework is interesting and novel.
2. The proposed OmniMixup framework is easy to follow.
3. The experiments are showing the effectiveness of the approach on two applications.
4. There are extensive theoretical analysis, which should be encouraged.

**Weaknesses:**

1. My major concern is that the operating range or limitation of this approach is not clear. Let me be clear: can this OmniMixup replace all other mixup approaches in applications? It seems not since this only deals with sample-level approaches (as shown in Sec. 3.2). There are other manifold space-based mixup approaches not discussed nor even compared. So, I’m confused the practical value of this approach: is it replacing existing approaches, or just a plugin for them?
2. The design of OmniMixup only supports sample-based approaches, while manifold mixup (Verma 2019) has been widely used in applications. It remains unclear whether this unified framework can include that type of approaches and how can they be compared.
3. The complexity of this approach is not discussed. From algorithm 1, it seems that this paper requires non-trivial optimization or iteration to select the best MPSD. However, this is not discussed in the paper. I’m not trying to be mean on this; but the efficiency should be very important in Mixup based approaches since the essence of Mixup family is simple and useful. Therefore, efficiency matters.
4. Experiments are weak, which is another major concern of mine. First of all, the comparison methods are not enough to show the effectiveness of this algorithm (manifold mixup, for instance, in not compared). Second, the datasets to perform image classification are not enough. Given that Mixup has been widely used in image domain, Cifar-100 is not enough. Authors should include more datasets and methods for comparisons.

**Questions:**

See weakness section.

---

> ### Author Response · Authors · 2023-11-13
> **Response to Reviewer sgeX**
>
> We thank the reviewer for their effort in the review and their constructive and valuable comments. Our responses to the questions are as below:
>
> **W1: My major concern is that the operating range or limitation of this approach is not clear. Let me be clear: can this OmniMixup replace all other mixup approaches in applications? It seems not since this only deals with sample-level approaches (as shown in Sec. 3.2). There are other manifold space-based mixup approaches not discussed nor even compared. So, I’m confused the practical value of this approach: is it replacing existing approaches, or just a plugin for them?**
>
> **RW1:** We argue that many popular mixup approaches are all within the line of works where modality features are exploited to increase the granularity of mixup. In this case, the usage of these approaches may be limited as the adaptation of them to text or graph data is not straightforward. Hence, we propose to modify mixup from another perspective which is shared by most of the mixup algorithms.  AdaMixUp, where the distribution of $\lambda$ is modified and ManifoldMixup, where the latent space of mixup is modified, are two lines of work in this perspective. As another line of work from this perspective, we propose to generalize the MPSDs. However, we argue that OmniMixup is flexible enough and integrable with either mixup approaches that exploit the modality-specific feature (e.g., CutMix) or more general approaches like AdaMixUp and ManifoldMixup. Therefore, OmniMixup is not used to replace the existing approach, but to generalize the original mixup process.
>
> **W2: The design of OmniMixup only supports sample-based approaches, while manifold mixup (Verma 2019) has been widely used in applications. It remains unclear whether this unified framework can include that type of approaches and how they can be compared.**
>
> **RW2:** As discussed in Response to Q1, OmniMixup and ManifoldMixup belong to different lines of work. Therefore, we mainly focus on the baselines where OmniMixup is generalized from.
>
> **W3: The complexity of this approach is not discussed. From algorithm 1, it seems that this paper requires non-trivial optimization or iteration to select the best MPSD. However, this is not discussed in the paper. I’m not trying to be mean on this; but the efficiency should be very important in Mixup based approaches since the essence of the Mixup family is simple and useful. Therefore, efficiency matters.**
>
> **RW3:** As shown in Sec. 3.4, one of the reasons the OmniEval framework is proposed is because we want to reduce the computational cost of MPSD searching. If we run OmniMixup multiple times with different MPSDs, except being computational costly, the comparisons among M-Score itself will also become completely meaningless as we have better evaluation metrics (e.g., F-1/Accuracy) for comparison. In contrast, with the OmniEval proposed, we are able run ERM only once and quickly calculate MPSDs and then compare. Note that the time cost of calculating M-Score for MPSDs are seconds-level, which is ignorable compared to training a model.
>
> **W4: Experiments are weak, which is another major concern of mine. First of all, the comparison methods are not enough to show the effectiveness of this algorithm (manifold mixup, for instance, in not compared). Second, the datasets to perform image classification are not enough. Given that Mixup has been widely used in image domain, Cifar-100 is not enough. Authors should include more datasets and methods for comparisons.**
>
> **RW4:** In terms of baselines, we believe that popular mixup approaches like CutMix, PuzzleMix all belong to a line of research that uses modality features to increase the granularity of mixup. However, the use of these approaches may be limited in this case, as adapting them to text or graph data is not straightforward. Hence, we propose to modify mixup from another perspective which is shared by most of the mixup algorithms.  ManifoldMixup is one line of work in this perspective where the mixing latent space is modified. As another line of work from this perspective, we propose to generalize the MPSDs. Therefore, we mainly focus on the baselines where OmniMixup is generalized from. However, note that a further modification on OmniMixup will definitely make it integrable with either CutMix/PuzzleMix or ManifoldMixup. We will add these experiments in our modified version of paper.
>
> In terms of experiments, thank you for your valuable feedback, we will add more experiments to elaborate the effectiveness of OmniMixup.

---

### Official Review · Reviewer_39U8 · 2023-10-24

**Soundness:** 1 poor
**Presentation:** 1 poor
**Contribution:** 2 fair
**Rating:** 3
**Confidence:** 3

**Summary:**

This paper introduces a mixup-based data augmentation technique that can be universally applied across various data modalities. While it extends the manifold mixup, which blends data in the latent space learned by the model rather than a specific data space, it employs a generalized form of the mixup methodology that encompasses a range of mixup-based algorithms. This generalized mixup is formalized, and its objective function's generalization bound is derived from the Rademacher complexity perspective. Building on this foundation, the paper proposes the OmniMixup methodology, which utilizes mixup pairs optimized for the derived bound in data augmentation. The introduced method is applied in both the vision domain and the graph data domain for molecule modeling, demonstrating improved performance compared to baseline models.

**Strengths:**

- The paper introduces OmniMixup, a generalized form of the mixup methodology that encompasses various mixup algorithms. It mathematically demonstrates that other mixup algorithms can be reduced to specific instances of OmniMixup.
- The generalization bound for the objective function of OmniMixup is derived using Rademacher complexity.
- An algorithm is presented to optimize the derived generalization bound, and a method to obtain the optimal MPSD is proposed.
- Based on the measured M-score, the authors select suitable latent feature acquisition models and mixup algorithms across diverse modalities and apply the proposed methodology.

**Weaknesses:**

- While the proposed OmniMixup is understood to theoretically sample data pairs optimized for the generalization bound, considering it only shows a slight performance improvement compared to mixup and isn't compared to recently proposed methodologies, the provided experimental results seem insufficient in showcasing its true effectiveness.
- Although the paper claims that OmniMixup can be applied across various modalities, the experiments presented are limited to low-resolution images and molecule graph data datasets. This restricts the ability to ascertain OmniMixup's broad applicability across different modalities.
- The motivation behind the design choice of MPSD in each modality is not adequately explained.
- It appears that a cost is associated with obtaining the optimized MPSD for mixup application. If this cost is significant and the performance improvement over other mixup methodologies is marginal, there might be limited incentive to employ OmniMixup.

**Questions:**

- Are there experimental results for other image datasets, and if so, how do they compare with existing mixup methodologies? Specifically, are there results for datasets that display relatively low performance during standard training, especially those with high resolution or various types of noise? The outcomes presented in the paper indicate only modest performance improvements when compared to baseline mixup results, making it challenging to discern its effectiveness. It would be better to show the experiments for little harder datasets to demonstrate the data augmentation effect of the proposed method.
- Has the method been tested on datasets of different modalities, e.g., NLP? Given that the mixup is applied in the latent space, it seems feasible to conduct experiments across various modalities.
- Are there results for other design choices of OmniMixup apart from the one used in the paper? Given OmniMixup's highly generalized representation, an ablation study exploring various design choices seems warranted.
- Is there a cost analysis experiment for executing the OmniEval algorithm? While the cost might vary based on the model choice for deriving features, if the performance justifies the incurred cost, it might be worthwhile to consider the methodology's application.

- Some notations in the paper could benefit from further clarification.
- The experimental descriptions are unclear. I interpreted from Figure 1 that higher performance corresponds to a lower M-score, but it's ambiguous what "performance" specifically refers to and how it's represented in quartiles.
- Numerous supplementary experiments appear necessary. If the aforementioned issues are addressed, I anticipate that the methodology could be a promising approach for data augmentation across various modalities, including the vision domain.

---

> ### Author Response · Authors · 2023-11-13
> **Response to Reviewer 39U8**
>
> We thank the reviewer for their effort in the review and their constructive, valuable and detailed comments. Our responses to the questions are as below:
>
> **W1: While the proposed OmniMixup is understood to theoretically sample data pairs optimized for the generalization bound, considering it only shows a slight performance improvement compared to mixup and isn't compared to recently proposed methodologies, the provided experimental results seem insufficient in showcasing its true effectiveness.**
>
> **RW1:** For performance, we consider that OmniMixup can consistently surpass the ERM and Mixup approach across diverse tasks in image and molecule classification. This aligns with our theoretical result for bounding the generalization ability of the model.
>
> **W2: Although the paper claims that OmniMixup can be applied across various modalities, the experiments presented are limited to low-resolution images and molecule graph data datasets. This restricts the ability to ascertain OmniMixup's broad applicability across different modalities.**
>
> **RW2:** Thank you for your valuable feedback! We will add more experiments for text, and high-resolution images classification tasks as you mentioned to further justify the effectiveness of OmniMixup.
>
> **W3: The motivation behind the design choice of MPSD in each modality is not adequately explained.**
>
> **RW3:** The reason we choose CLIP and fingerprint to construct MPSDs for both images and molecules is that the generated latent features can well represent the image / molecules and therefore it is convenient for us to show the MPSD-searching-and-training process utilizing the proposed OmniEval framework. And the reason we do not use the latent features during training to construct MPSDs is that the MPSDs will change in every step, which is not aligned with the theoretical results. Note that as the OmniEval framework only requires a set of MPSDs to select the best one within the set, any MPSD that is validly defined can be added into this set for searching. We will add more experiments to elaborate this issue in our modified version.
>
> **W4: It appears that a cost is associated with obtaining the optimized MPSD for mixup application. If this cost is significant and the performance improvement over other mixup methodologies is marginal, there might be limited incentive to employ OmniMixup.**
>
> **RW4:** As shown in Sec. 3.4, one of the reasons the OmniEval framework is proposed is because we want to reduce the computational cost of MPSD searching. If we run OmniMixup multiple times with different MPSDs, except being computational costly, the comparisons among M-Score itself will also become completely meaningless as we have better evaluation metrics (e.g., F-1/Accuracy) for comparison. In contrast, with the OmniEval proposed, we are able run ERM only once and quickly calculate MPSDs and then compare. Note that the time cost of calculating M-Score for MPSDs are seconds-level, which is ignorable compared to training a model.

---

> > ### Author Response · Authors · 2023-11-13
> > **Response to Reviewer 39U8 (continued)**
> >
> > **Q1: Are there experimental results for other image datasets, and if so, how do they compare with existing mixup methodologies? Specifically, are there results for datasets that display relatively low performance during standard training, especially those with high resolution or various types of noise? The outcomes presented in the paper indicate only modest performance improvements when compared to baseline mixup results, making it challenging to discern its effectiveness. It would be better to show the experiments for little harder datasets to demonstrate the data augmentation effect of the proposed method.**
> >
> > **Q2: Has the method been tested on datasets of different modalities, e.g., NLP? Given that the mixup is applied in the latent space, it seems feasible to conduct experiments across various modalities.**
> >
> > **RQ1, RQ2:** Please refer to **RW2**.
> >
> > **Q3: Are there results for other design choices of OmniMixup apart from the one used in the paper? Given OmniMixup's highly generalized representation, an ablation study exploring various design choices seems warranted.**
> >
> > **RQ3:** Please refer to **RW3**.
> >
> > **Q4: Is there a cost analysis experiment for executing the OmniEval algorithm? While the cost might vary based on the model choice for deriving features, if the performance justifies the incurred cost, it might be worthwhile to consider the methodology's application.**
> >
> > **RQ4:** In terms of the mixup-based training, the computational cost is similar to the ERM algorithm. In terms of the computational cost, please refer to **RW4**.
> >
> > **Q5: Some notations in the paper could benefit from further clarification.**
> >
> > **RQ5:** Thank you for your advice, we will carefully double check the notations in the paper and revise them to make the paper clearer and easier to follow. If you have found any that are needed to be modified, we will appreciate it if you point them out.
> >
> > **Q6: The experimental descriptions are unclear. I interpreted from Figure 1 that higher performance corresponds to a lower M-score, but it's ambiguous what "performance" specifically refers to and how it's represented in quartiles.**
> >
> > **RQ6:** To make it clearer to the reader, we have updated the box plots in Figure 1 to scatter plots in the updated version of paper, which have been uploaded in OpenReview. The updated plots show that there is a linear association between M-Score and the performance of MPSDs, and all of the associations are significant with a p-value less than 0.05.
> >
> > **Q7: Numerous supplementary experiments appear necessary. If the aforementioned issues are addressed, I anticipate that the methodology could be a promising approach for data augmentation across various modalities, including the vision domain.**
> >
> > **RQ7:** Thank you for your insightful feedback. We will add more comprehensive empirical study toward the proposed algorithm in our revised version of paper.

---

### Official Review · Reviewer_CqYZ · 2023-10-31

**Soundness:** 2 fair
**Presentation:** 3 good
**Contribution:** 2 fair
**Rating:** 5
**Confidence:** 3

**Summary:**

The authors first propose OmniMixup framework that works with arbitrary Mixing-Pair Sampling Distribution (MPSD). Essentially, it determines how to choose a pair of samples to produce virtual examples that are not out-of-distribution. Then they find that the Mahalanobis distance (M-Score) is highly correlated to the performance of given MPSD so propose OmniEval for choosing optimal MPSD before training.

**Strengths:**

1. Overall, the paper is well-written, and its ideas are presented in a clear and easily understandable manner for readers to follow.
2. The paper introduces OmniMixup, a generalized framework that extends the concept of Mixup and addresses the limitations of modality-specific approaches. It introduces Mixing-Pair Sampling Distribution (MPSD) and provides a theoretical analysis framework for evaluating the generalization ability of different approaches. To the best of my knowledge, the OmniMixup framework is novel.
3. The theoretical aspect of the paper appears to be solid.

**Weaknesses:**

Overall, my main concern is that the experimental results do not sufficiently demonstrate OmniMixup's advantage in terms of model performance. For instance, the advantage shown in Table 1 is less than 0.6%, and the advantage shown in Table 2 is not statistically significant. The limited empirical evidence diminishes the significance of formulating the framework.

**Questions:**

1. Section 2.3 of [1] presents a case that demonstrates how mixup may lead to a classifier that does not minimize the empirical loss on the data. I am interested to know if OmniMixup can avoid this issue.
2. Is it possible to utilize a pre-trained model in the first step of Algorithm 1, eliminating the need for training with ERM?
3. I noticed that Table 1 does not include standard deviations like Table 2. Since the improvement of OmniMixup on the CIFAR dataset is marginal, for example, only 0.2%, which could potentially be influenced by randomness. It would be helpful for the authors to include standard deviations, similar to Table 2.
4. The experiments section should include more baselines, such as Local-Mix and C-Mixup. This would help demonstrate if the optimal pair sampling distribution in OmniMixup outperforms strategies designed by experts.
5. Figure 1 is not clear enough for me. To demonstrate the relationship between the estimated M-Score and their respective model performances, it might be more effective to provide a scatter plot or plot their ranks similar to Figure 8 in [2]. Additionally, calculating the Kendall-tau correlation could further justify your points.

---

> ### Author Response · Authors · 2023-11-13
> **Response to Reviewer CqYZ**
>
> We thank the reviewer for their review and their constructive and valuable comments. Our responses to the questions are as below:
>
> **W1: Overall, my main concern is that the experimental results do not sufficiently demonstrate OmniMixup's advantage in terms of model performance. For instance, the advantage shown in Table 1 is less than 0.6%, and the advantage shown in Table 2 is not statistically significant. The limited empirical evidence diminishes the significance of formulating the framework.**
>
> **RW1:** For performance, we consider that OmniMixup can consistently surpass the ERM and Mixup approach across diverse tasks in image and molecule classification. This aligns with our theoretical result for bounding the generalization ability of the model.
>
> **Q1: Section 2.3 of [1] presents a case that demonstrates how mixup may lead to a classifier that does not minimize the empirical loss on the data. I am interested to know if OmniMixup can avoid this issue.**
>
> **RQ1:** Could you please provide the name of the paper you refer to so that we could have a further discussion on this question? Thank you!
>
> **Q2: Is it possible to utilize a pre-trained model in the first step of Algorithm 1, eliminating the need for training with ERM?**
>
> **RQ2:** This is a great question and we are happy to discuss. The reason we can use an ERM model instead is that we made a concession to assume that parameters of a model trained with ERM may also fall into the set shown in Assumption 3 if we make the $\gamma$ large enough. To this end, we are able to directly use $x$ generated by the ERM model to estimate any given MPSD within a friendly computational budget. While note that a poor performance of ERM model may make the upper bound looser. Hence, in terms of the pre-trained model, we are able to utilize it as long as it has a strong performance already in zero-shot setting. Otherwise the upper bound may be too loose and therefore unreliable.
>
> **Q3: I noticed that Table 1 does not include standard deviations like Table 2. Since the improvement of OmniMixup on the CIFAR dataset is marginal, for example, only 0.2%, which could potentially be influenced by randomness. It would be helpful for the authors to include standard deviations, similar to Table 2.**
>
> **RQ3:** Thank you for your valuable advice, we will add standard deviation for image classification benchmark in our revised version.
>
> **Q4: The experiments section should include more baselines, such as Local-Mix and C-Mixup. This would help demonstrate if the optimal pair sampling distribution in OmniMixup outperforms strategies designed by experts.**
>
> **RQ4:** Thank you for the advice. In the current benchmark, we will try to re-implement the experiment in their benchmark and make comparisons to OmniMixup in our revised version of paper.
>
> **Q5: Figure 1 is not clear enough for me. To demonstrate the relationship between the estimated M-Score and their respective model performances, it might be more effective to provide a scatter plot or plot their ranks similar to Figure 8 in [2]. Additionally, calculating the Kendall-tau correlation could further justify your points.**
>
> **RQ5:** Thank you for your suggestion. We have added a scatter plot to our updated paper. The updated plots show that there is a linear association between M-Score and the performance of MPSDs, and all of the associations are significant with a p-value less than 0.05.

---

> > ### Comment · Reviewer_CqYZ · 2023-11-22
> > **Thanks for your rebuttal**
> >
> > I think the author did not address my main concern. I believe that solid experimental results are crucial for this paper, so I strongly recommend the author to compare various variants of mixup that currently exist and demonstrate the improvements brought by introducing complex MPSD. I apologize for the missing citation.
> >
> > [1] Chidambaram, M., Wang, X., Hu, Y., Wu, C., & Ge, R. (2021, October). Towards Understanding the Data Dependency of Mixup-style Training. In International Conference on Learning Representations (Spotlight).
> >
> > [2] Yang, A., Esperança, P. M., & Carlucci, F. M. (2019, September). NAS evaluation is frustratingly hard. In International Conference on Learning Representations.

---

### Official Review · Reviewer_AcyS · 2023-10-31

**Soundness:** 2 fair
**Presentation:** 1 poor
**Contribution:** 1 poor
**Rating:** 1
**Confidence:** 4

**Summary:**

The way the paper is written is quite confusing. From my current understanding of it, the authors formalise the mechanism of the choice of the samples to be mixed as a distribution that can be automatically selected through an algorithm that considers the mahalanobis score. They frame previously existing mixing techniques as a subcase of their own, derive some bounds and provide some experiments that are supposed to exhibit the superiority of the proposed technique.

**Strengths:**

- The paper tries to provide a theoretical framework that drives the selection of the samples to be mixed.

**Weaknesses:**

- The motivation of the paper is confusing. The authors first mention that mixup is hard to apply across different modalities, but it's not clear what the authors are doing specifically about it. This is even more confusing when mixup techniques performed in latent space are brought up, yet I seem to understand the authors did not use those techniques.
- It is not clear why the authors do not compare Omnimixup with the extensively available mixup variants. For instance, sticking to image processing: why is there not a comparison with CutMix [1], RegMixup [2], PuzzleMix [3], AugMix [4] just to mention a few. Similarly other baselines of Mixup variants for molecular data should be considered.
- From the image classification experiments, it looks like the models of the baselines have not been properly trained. For instance, in [2] the numbers reported for Mixup and ERM on WideResNet-28-10 are higher.
- The paper often refers to manifold intrusion, which is a questionable concept introduced in the Mixup literature but not adequately studied or quantified. It is not really clear how Omnimixup either proves its existence or is doing something about it.
- If access to CLIP is required to perform Omnimixup on images, then one would be better off using CLIP directly as a classifier and fine-tune it/use smarter strategies to use it. What's the point of even bothering about Omnimixup to get marginal accuracy improvements on toy datasets like CIFAR-10/100?
- An algorithm box to exemplify omnimixup would help.
- The presentation is extremely chaotic and confusing, some terms are used without being properly defined, and one needs to jump back and forth between different parts of the paper to try to figure out what's going on.


[1] https://arxiv.org/abs/1905.04899
[2] https://arxiv.org/pdf/2206.14502.pdf
[3] https://arxiv.org/abs/2009.06962
[4] https://arxiv.org/pdf/1912.02781.pdf

**Questions:**

- Is the mixing performed in input space or latent space?
- The search procedure seems to come at a high computational expense. Could the authors precisely report and discuss the cost of searching both theoretically and empirically? If this cost is inferior to training few models with ERM, even ensembling should be considered as a baseline.

---

> ### Author Response · Authors · 2023-11-13
> **Response to Reviewer AcyS**
>
> We thank the reviewer for their review and their constructive and valuable comments. Our responses to the questions are as below:
>
> **W1: The motivation of the paper is confusing. The authors first mention that mixup is hard to apply across different modalities, but it's not clear what the authors are doing specifically about it. This is even more confusing when mixup techniques performed in latent space are brought up, yet I seem to understand the authors did not use those techniques.**
>
> **RW1**: We argue that the motivation of our proposed mixup approach is that instead of utilizing features that are unique for a particular modality (e.g., pixels in image, word in text, and node in graph) to increase the granularity of mix-up, we proposed to modify the shared components of the mix-up algorithm that are consistent across all modalities, which is the mix-up pair matching process within OmniMixup. In this case, OmniMixup is readily applicable even to data involving novel modalities that only have limited prior exploration in mixup.
>
> **W2: It is not clear why the authors do not compare OmniMixup with the extensively available mixup variants. For instance, sticking to image processing: why is there not a comparison with CutMix [1], RegMixup [2], PuzzleMix [3], AugMix [4] just to mention a few. Similarly other baselines of Mixup variants for molecular data should be considered.**
>
> **RW2:** As discussed in **RW1**, we consider that popular mixup approaches like CutMix, RegMixup, PuzzleMix, and AugMix all belong to a line of research that uses modality features to increase the granularity of mixup. However, the use of these approaches may be limited in this case, as adapting them to text or graph data is not straightforward. Hence, we propose to modify mixup from another perspective which is shared by most of the mixup algorithms. AdaMixup is one line of work in this perspective where the distribution of $\lambda$ is modified. As another line of work from this perspective, we propose to generalize the MPSDs. Therefore, we mainly focus on the baselines where OmniMixup is generalized from. However, note that a further modification on OmniMixup will definitely make it integrable with either CutMix/PuzzleMix or AdaMixup. We will add these experiments in our modified version of paper.
>
> **W3: From the image classification experiments, it looks like the models of the baselines have not been properly trained. For instance, in [2] the numbers reported for Mixup and ERM on WideResNet-28-10 are higher.**
>
> **RW3:** We want to clarify that the experiments are fair as both baselines and the proposed approach on image classification are implemented based on the source code of Mixup. While we acknowledge that there are still performance gaps between our re-implemented baselines and the one reported in the original Mixup paper, we are currently trying our best to resolve this issue.
>
> **W4: The paper often refers to manifold intrusion, which is a questionable concept introduced in the Mixup literature but not adequately studied or quantified. It is not really clear how OmniMixup either proves its existence or is doing something about it.**
>
> **RW4:** We want to clarify that we only refer to **manifold intrusion** in introduction and related work sections to review what they stated in their original papers. We **completely agree** with your opinion that manifold intrusion is a questionable concept and vague in mixup literatures. Therefore we did not elaborate our approach in this way in our paper.
>
> **W5: If access to CLIP is required to perform OmniMixup on images, then one would be better off using CLIP directly as a classifier and fine-tune it/use smarter strategies to use it. What's the point of even bothering about OmniMixup to get marginal accuracy improvements on toy datasets like CIFAR-10/100?**
>
> **RW5:** We want to clarify that since the OmniEval framework only requires a set of MPSDs to select the best one from the set, any validly defined MPSD can be added to this set for searching. Therefore, it is not necessary to have access to CLIP to construct MPSD when performing OmniMixup on images. We chose CLIP because its generated latent features can accurately represent the image, which makes it convenient for us to demonstrate the MPSD-searching-and-training process using the proposed OmniEval framework. And the reason we do not use the latent features during training to construct MPSDs is that the MPSDs will change in every step, which is not aligned with the theoretical results.

---

> ### Author Response · Authors · 2023-11-13
> **Response to Reviewer AcyS (continued)**
>
> **W6: An algorithm box to exemplify OmniMixup would help.**
>
> **RW6:** Thanks for your advice. Yes I agree this will be really helpful for readers. We will update an algorithm box about OmniMixup in our paper later.
>
> **W7: The presentation is extremely chaotic and confusing, some terms are used without being properly defined, and one needs to jump back and forth between different parts of the paper to try to figure out what's going on.**
>
> **RW7:** We appreciate your feedback on the structure of the paper. We will improve it in our modified version. We would be grateful if you could point out any confusing parts in the paper.
>
> **Q1: Is the mixing performed in input space or latent space?**
>
> **RQ1:** The OmniMixup is mainly performed in latent space. While note that for images we are also able to perform in input space as the mixing of two images is straightforward, unlike text and graph.
>
> **Q2: The search procedure seems to come at a high computational expense. Could the authors precisely report and discuss the cost of searching both theoretically and empirically? If this cost is inferior to training a few models with ERM, even ensembling should be considered as a baseline.**
>
> **RQ2:** This is a really good question. In fact, the OmniEval framework was proposed to reduce the computational cost of MPSD searching, as shown in Section 3.4. If we run OmniMixup multiple times with different MPSDs, it would be computationally expensive and the comparisons between M-Scores would be meaningless, as we have better evaluation metrics (e.g., F-1/Accuracy) for comparison. In contrast, with the OmniEval proposed, we are able run ERM only once and quickly calculate MPSDs and then compare. Note that the time cost of calculating M-Score for MPSDs are seconds-level, which is ignorable compared to training a model.
>
> In terms of the validity of using ERM trained model for MPSDs-searching, we recall that Assumption 3 requires the upper bound to hold only if the model falls within the set $W_\gamma$, which is a set of all parameters that lead to a well-trained model (i.e., has small main and regularization loss) under OmniMixup. We make the assumption that the parameters of a model trained with ERM may also fall into this set if we make the $\gamma$ reasonably large enough. This allows us to directly use $x$ generated by the ERM model to estimate any given MPSD.

---

> > ### Comment · Reviewer_AcyS · 2023-11-21
> > **Thanks for your rebuttal**
> >
> > W1: Thanks, it is clearer. However, for the approach to be useful maybe it would be better to focus on those modalities for which SOTA improvements on mixup have not been produced because of the problem you point at.
> >
> > W2: As mentioned in the previous point, it could help to focus on modalities for which vicinal distributions are hard to design to make the paper stronger. Where these vicinal distributions are easy to identify, then the method should either be composable in a useful way with existing methods or should outperform them. I did not notice the upload of a revision with additional experiments though.
> >
> > W3: This is a big  concern. I would also remark that several papers and frameworks have implemented Mixup themselves. It could probably help you reproducing high-performance results reusing publicly available code from other sources.
> >
> > W4: Ok
> >
> > W5: If you could show CLIP can be replaced with a less powerful model it would help to alleviate the issue. If access to CLIP is assumed, there are several more sophisticated things that can be done with it that would probably significantly outperform OmniMixup.
> >
> > W6 onwards: Ok

---

### Official Review · Reviewer_wmTG · 2023-11-01

**Soundness:** 3 good
**Presentation:** 3 good
**Contribution:** 2 fair
**Rating:** 5
**Confidence:** 4

**Summary:**

This paper studies the Mixup technique and proposes to sample across different modalities to explore the generalization of Mixup training. Through extensive theoretical analysis, the authors find out that the Mahalanobis distance is essential for the capabilities of Mixup. By proposing a novel Mixing-Pair Sampling Distribution to summarize the most popular Mixup strategies, a uniform framework is introduced to provide a holistic understanding of Mixup. By conducting rigorous experiments, the performance of the proposed OmniMixup is carefully validated.

**Strengths:**

- This paper is well-written and easy to follow.
- The provided generalization bound is rigorously derived and provides a solid understanding of introducing the expected M-Score criteria.
- Through careful empirical validation, the performance OmniMixup is shown to surpass ERM and Mixup.

**Weaknesses:**

- First, could the author justify the significance of Mixup over multiple modalities? What is the basic motivation and novelty?
- It seems that the modalities are just sampled from image and text features provided by CLIP. However, it is questionable to call these two types of features multimodalities, because CLIP is a very strong model, and its feature spaces are perfectly aligned with each other. Is it possible that the proposed method can be applied to low-level multimodalities?
- Is there any intuitive understanding of the theoretical result? How is the direct comparison of the proposed generalization bound to vanilla Mixup and ERM?
- Is the hyperparameters $\tau$ and $\beta$ carefully selected?
- The experimental improvements are quite marginal, which again raises concern about the significance of proposing such a holistic framework. Moreover, the M-score is addressed in this paper, however, the comparison in Table 3 seems to challenge the claim: the M-scores of Mixup and OmniMixup are very close to each other.

**Questions:**

Please refer to the weaknesses.

**Details Of Ethics Concerns:**

No ethics concerns.

---

> ### Author Response · Authors · 2023-11-13
> **Response to Reviewer wmTG**
>
> We thank the reviewer for their review and their constructive and valuable comments. Our responses to the questions are as below:
>
> **W1: First, could the author justify the significance of Mixup over multiple modalities? What is the basic motivation and novelty?**
>
> **RW1:** We argue that the novelty of OmniMixup lies in the fact that it does not use features that are unique to a particular modality (e.g., pixels in an image, words in text, or nodes in a graph) to increase the granularity of mixup. Instead, we propose to modify the shared components of the mixup algorithm that are consistent across all modalities, which is the mixup pair matching process within OmniMixup. In this case, OmniMixup can be readily applied to data involving novel modalities that have only had limited prior exploration in mixup. Accordingly, we further theoretically analyze the OmniMixup method, and propose OmniEval to help search for the best MPSD for a given task.
>
> **W2: It seems that the modalities are just sampled from image and text features provided by CLIP. However, it is questionable to call these two types of features multimodalities, because CLIP is a very strong model, and its feature spaces are perfectly aligned with each other. Is it possible that the proposed method can be applied to low-level multimodalities?**
>
> **RW2:** We argue that CLIP for images and fingerprint for molecules are used for constructing the corresponding MPSDs only. The reason we do not use the latent features during training to construct MPSDs is that the MPSDs will change in every step, which is not aligned with the theoretical results.
>
> In terms of the multimodalities, we want to clarify that the reason we mention “modality” in the paper is to state the motivation of our approach as elaborated in response to W1. We did not claim that our approach is able to work well under multimodalities, we want to claim that compared to those mixup approaches that exploit the modality feature a lot, our approach can be much easier applied to tasks across different modalities without any further adaptation, as the vanilla mixup or ManifoldMixup.
>
> **W3: Is there any intuitive understanding of the theoretical result? How is the direct comparison of the proposed generalization bound to vanilla mixup and ERM?**
>
> **RW3:** As discussed in Sec 3.4, from Theorem 2, to improve the generalization ability of a given MPSD, we should control $\mathbb{E}[x^\top
>  \Sigma^{-1} x]$ (i.e., expected M-Score) to shrink the upper bound as much as possible. However, two challenges remain here:
>
> 1. We cannot calculate the expected M-Score because of the inaccessibility of the population. A straightforward solution is applied to tackle this issue: we calculate the sample mean of the expected M-Score using the training dataset.
>
> 2. Assumption 3 requires the upper bound to hold only if the model falls within the set $W_\gamma$ , which is a set of all parameters that lead to a well-trained model (i.e., has small main and regularization loss) under OmniMixup. In this case, we cannot calculate the estimated M-Score because we cannot access $x$ before we train the model. However, this makes the expected M-Score meaningless because we have better evaluation metrics once we have trained the model. Therefore, we make the assumption that the parameters of a model trained with ERM may also fall into this set if we make the $\gamma$ reasonably large enough. This allows us to directly use $x$ generated by the ERM model to estimate any given MPSD within a friendly computational budget.
> The addressing of the above challenges led to OmniEval. Given a set of candidate MPSD,  OmniEval can choose the best to fit the model with only one run under ERM.
>
> **W4: Are the hyperparameters beta and tau carefully selected?**
>
> **RW4:** We first created a list of beta and tau values, and then selected the one with the lowest estimated M-Score using the OmniEval framework. We then fixed these two hyperparameters and trained the model. It is important to note that we should carefully select beta and tau, and even more families of MPSDs, as we want to find the best MPSD to create the tightest upper bound in Theorem 2.

---

> > ### Author Response · Authors · 2023-11-13
> > **Response to Reviewer wmTG (continued)**
> >
> > **W5: The experimental improvements are quite marginal, which again raises concern about the significance of proposing such a holistic framework. Moreover, the M-score is addressed in this paper, however, the comparison in Table 3 seems to challenge the claim: the M-scores of Mixup and OmniMixup are very close to each other.**
> >
> > For performance, we consider that OmniMixup can consistently surpass the ERM and Mixup approach across diverse tasks in image and molecule classification. This aligns with our theoretical result for bounding the generalization ability of the model.
> >
> > For the estimated M-Score, note that it is meaningless to consider only the difference between two M-Scores. From Theorem 2, it is important to find out that when $L$ , $L_A$ , $\beta$ (this beta is different from the hyperparameter, sorry for the confusion, we have uploaded the modified version) is large, a small difference between M-Score will still significantly narrow the gap between the empirical loss and the population loss.

---

### Meta-Review · Area_Chair_p9sJ · 2023-12-08

**Metareview:**

This paper introduces OmniMixup, a new modification of the Mixup data augmentation technique that aims to mitigate overfitting in empirical risk minimization. Current mixup modifications are modality-specific, thus limiting their applicability across different modalities. OmniMixup generalizes prior work by introducing a Mixing-Pair Sampling Distribution (MPSD) and proposes to sample across different modalities. A key finding of the research is that the Mahalanobis distance (M-Score), derived from the sampling distribution, provides insights into the generalization capabilities of OmniMixup. This insight led to the development of OmniEval, an autonomous evaluation framework designed to identify the optimal sampling distribution.

However, the reviewers have various concerns about the novelty, contribution, experiments, and clarity of the current draft, leaving an unanimous agreement for rejection. So is the final decision.

**Justification For Why Not Higher Score:**

The reviewers unanimous reject the submission.

**Justification For Why Not Lower Score:**

N.A.

---

### Decision · Program_Chairs · 2024-01-16

Reject